# Biomarkers for Immune Checkpoint Inhibitors in Renal Cell Carcinoma

**DOI:** 10.3390/jcm12154987

**Published:** 2023-07-28

**Authors:** Spencer D. Martin, Ishmam Bhuiyan, Maryam Soleimani, Gang Wang

**Affiliations:** 1Department of Pathology and Laboratory Medicine, Faculty of Medicine, University of British Columbia, Vancouver, BC V5Z 1M9, Canada; spencer.martin@vch.ca; 2Faculty of Medicine, University of British Columbia, Vancouver, BC V6T 1Z3, Canada; ibhuiyan@student.ubc.ca; 3Division of Medical Oncology, Faculty of Medicine, University of British Columbia, Vancouver, BC V6T 1Z3, Canada; maryam.soleimani@bccancer.bc.ca; 4British Columbia Cancer Vancouver Centre, Vancouver, BC V5Z 4E6, Canada

**Keywords:** immune checkpoint inhibitor, renal cell carcinoma, biomarker, tumor mutation burden, T cell, PD-L1, PD1, VHL, PBRM-1, immunotherapy

## Abstract

Immune checkpoint inhibitor (ICI) therapy has revolutionized renal cell carcinoma treatment. Patients previously thought to be palliative now occasionally achieve complete cures from ICI. However, since immunotherapies stimulate the immune system to induce anti-tumor immunity, they often lead to adverse autoimmunity. Furthermore, some patients receive no benefit from ICI, thereby unnecessarily risking adverse events. In many tumor types, PD-L1 expression levels, immune infiltration, and tumor mutation burden predict the response to ICI and help inform clinical decision making to better target ICI to patients most likely to experience benefits. Unfortunately, renal cell carcinoma is an outlier, as these biomarkers fail to discriminate between positive and negative responses to ICI therapy. Emerging biomarkers such as gene expression profiles and the loss of pro-angiogenic proteins VHL and PBRM-1 show promise for identifying renal cell carcinoma cases likely to respond to ICI. This review provides an overview of the mechanistic underpinnings of different biomarkers and describes the theoretical rationale for their use. We discuss the effectiveness of each biomarker in renal cell carcinoma and other cancer types, and we introduce novel biomarkers that have demonstrated some promise in clinical trials.

## 1. Introduction

Renal cell carcinoma (RCC) is a deadly disease, with 430,000 new cases diagnosed and 180,000 deaths reported worldwide in 2020 [1] and accounting for 2–3% of all malignant tumors in adults [2]. Clear cell RCC (ccRCC) accounts for between 75–80% of all RCC cases, while the remainder comprises several subtypes, including papillary RCC, chromophobe RCC, medullary RCC, collecting duct RCC, Xp11 and t(6,11) translocation RCCs, and others [3]. Approximately 70% of patients present with localized disease, either organ-confined or locally advanced, for which surgery is the primary curative strategy. Nonetheless, about 40% of these patients experience cancer recurrence and/or distant metastases post-surgery. Of the RCC patients that present with metastatic disease, the prognosis is dire, with an estimated 5-year survival rate of 15% [4]. Metastatic RCC remains incurable despite an improved understanding of the mechanistic and genetic basis of disease. Encouragingly, efforts to better characterize RCC have led to an increased number of treatment options, including tyrosine kinase inhibitors (TKIs), angiogenesis inhibitors, and immune checkpoint inhibitors (ICI) [4]. In the past decade, treatments with immunotherapeutic agents, alone or in multiple combinations, have dramatically increased the armamentarium against RCC, particularly ccRCC [5].

Immune checkpoint ligands on tumor cells or antigen-presenting cells (APC) bind to immune checkpoint molecules on activated T cells, leading to T cell anergy and inhibition of anti-tumor immunity. Immune checkpoint inhibitors (ICIs) are monoclonal antibodies that bind and block this interaction, releasing the brakes on anti-tumor immunity and allowing continued T-cell attack on the tumor [6]. The first approved ICIs block the CTLA-4—CD28 and PD-1—PD-L1 axes, with subsequent studies investigating the blockage of TIM3, LAG3, and others [7]. In RCC, several clinical trials show that ICIs induce potent anti-tumor responses, but although remarkably effective for some patients, ICIs fail to induce tumor regression in other patients [6,7]. The Food and Drug Administration and Health Canada approved first-line ICI-based regimens for advanced RCC include ipilimumab plus nivolumab, pembrolizumab plus axitinib, cabozantinib plus nivolumab, and lenvatinib plus pembrolizumab. Treatment is allocated based on International Metastatic Renal Cell Carcinoma Database Consortium (IMDC) categorization, the presence of sarcomatoid pathology, comorbid conditions, symptomatic disease burden, and patient preferences. All these regimens have demonstrated overall survival (OS) and progression-free survival (PFS) benefits, as well as complete responses in the range of 12–18% [8,9,10,11]. Unfortunately, anti-tumor immunotherapies often instigate autoimmune adverse events, including autoimmune hepatitis, thyroiditis, hypophysitis, colitis, interstitial lung disease, and many others [12]. To date, biomarkers that predict response to ICIs in advanced RCC are under intense investigation, yet none are approved for clinical use.

Identifying the patients best-suited for ICI therapy may drastically improve survival while limiting exposure to autoimmune adverse events. In this review, we comprehensively assess potential biomarkers that predict responses to immune checkpoint inhibitors in RCC. We provide updated assessments of the tumor mutation burden, tumor immune infiltration, immune checkpoint molecules, gene expression profiles, angiogenic pathways, and tumor intrinsic genetic and proteomic factors as biomarkers. We present a unique combination of the basic science that mechanistically underpins the use of each biomarker and the phase III clinical trial evidence that evaluates each biomarker in RCC. We further present evolving research to combine various biomarkers to predict responses to ICI. This review provides a comprehensive resource for understanding the rationale behind biomarkers in ICI and is meant to encourage the further development and validation of biomarkers for ICI in RCC.

## 2. A Brier Primer on Anti-Tumor Immunity

During the process of tumorigenesis, cancer cells develop tumor-specific antigens potentially recognizable as different or “foreign” by T cells of the immune system. Antigen-presenting cells (APC) phagocytose dying tumor cells, migrate to nearby lymph nodes, and process tumor proteins into peptides, a small subset of which bind to and are presented on the major histocompatibility complex (MHC). An APC can activate a naïve T cell if the T cell receptor (TCR) of a naïve T cell binds the peptide–MHC complex with sufficient affinity and the APC expresses the appropriate co-stimulatory molecules. The activated T cell proliferates and differentiates into an army of clonal cytotoxic CD8+ T cells (CTLs) or helper CD4+ T cells (THs), with each T cell expressing an identical TCR and recognizing the same peptide–MHC (or “antigen”). The armies of CTL and TH migrate to inflamed tissue and interrogate the peptide–MHC on cells within the tissue. If the T cells bind a cognate antigen presented on a cell, the T cells release cytokines and cytotoxic granules, causing the target cell to die (reviewed in [13]).

In an ideal situation, the army of CTLs moves from tumor cell to tumor cell, killing each cell sequentially and eradicating the entire tumor. However, negative feedback mechanisms have evolved to prevent overwhelming immune/autoimmune responses, and tumors often co-opt these mechanisms to escape anti-tumor immunity. Multiple cell types inhibit CTL activity within the tumor microenvironment (TME). For example, M2-polarized (wound-healing) tumor-associated macrophages (TAM) induce CTL anergy and recruit regulatory T cells (Tregs) to the TME [14]. Tregs, directly and indirectly, inhibit CTLs, polarize TAMs to the M2 phenotype, and inhibit antigen presentation [14,15]. Myeloid-derived suppressor cells (MDSCs) use contact-dependent and -independent mechanisms to inhibit the anti-tumor immune response [16]. In addition to combating these tumor-associated anti-immune cell types, anti-tumor immunity must overcome the following immune escape mechanisms: (a) tumors may fail to express antigens on the cell surface [17,18]; (b) tumor cells may prevent T cells from entering the tumor microenvironment [19]; (c) tumors may induce upregulation of immune checkpoint molecules on T cells [13,20]; (d) tumors and associated cells may secrete cytokines that inhibit T cells [21]; (e) tumors may upregulate immune checkpoint ligands that prevent CTL from killing them [22]. Each tumor may evolve one or several mechanisms to evade anti-tumor immunity, and mechanistic insights into immune evasion provide the rationale behind potential biomarkers of successful responses to ICI.

## 3. Tumor Mutation Burden

The spectrum of antigens on tumors cells is a key determinant of an effective anti-tumor response to ICI. Several lines of evidence implicate neoantigens—tumor-specific mutated peptides recognized by T cells—as key targets of anti-tumor immunity. First, they are, by definition, tumor-specific; therefore, T cells recognizing mutations escape negative thymic selection and often have very high affinities for their cognate neoantigen [23,24]. Indeed, mutation-specific T cells have a higher affinity for cognate epitopes compared to those from overexpressed antigens, cancer testis antigens, and other tumor-associated antigens [25]. Second, studies interrogating the repertoire of intratumoral T cells identified neoantigens as the most abundant antigen class [26,27]. Third, a robust T cell response targeted to a single neoantigen is capable of regressing established human metastatic pancreatic and colorectal tumors [28,29]. Unfortunately, only a small subset of tumor mutations are translated, processed and presented on MHC molecules, and recognized by T cells as neoantigens [17,30,31]. Thus, an increasing tumor mutation burden (TMB) increases the likelihood that a tumor presents increased numbers of neoantigens.

TMB measures the total number of somatic mutations in a given tumor, and multiple technical factors can influence TMB values [32]. In most studies investigating TMB, non-synonymous single-nucleotide variants (SNV) make up the majority of the mutations, while small insertions and deletions (indels) make up a minor fraction [32,33]. Synonymous variants may also be included in the TMB count to identify highly mutated tumors, though the immune system is unlikely to recognize these [32]. Ideally, TMB is assessed using whole-genome sequencing data, but recent studies show that tumor gene panel sequencing effectively estimates the TMB provided that at least 667 kb of DNA is sequenced [34]; however, inconsistent TMB estimates have been identified among different panels [35]. Sequencing circulating tumor DNA (ctDNA) can also estimate TMB, though the source of DNA may fail to represent the majority of the tumor bulk due to intratumoral heterogeneity and contaminating somatic mutations in hematopoietic cells [36].

The likelihood of encoding a neoantigen is different among various categories of mutations. For example, single-nucleotide variants (SNVs) generally result in a single amino acid substitution in the resultant protein, meaning the potential neoantigen is only slightly different from the self. In contrast, insertion and deletion (indel) mutations may result in a variety of changes to the protein [37] (Figure 1). While in-frame indels may insert or delete a single or a few amino acids and be similar to self-peptide sequences, indels often result in frameshift mutations and novel open reading frames, a series of amino acids completely different from any self-peptide [33]. They may also lead to fusion proteins, where the amino acid sequence substantially differs from any self-peptide. A series of amino acids drastically different from self-increases the possibility of generating a neoantigen compared to single amino acid substitutions (Figure 1) [38,39]. Indeed, a pan-cancer analysis of over 300,000 SNVs and 20,000 indels determined that indel frameshift mutations resulted in three-fold more predicted high-affinity neoantigens per mutation compared to SNVs [39]. Furthermore, mismatch repair deficient tumors containing abundant indel frameshift mutations are highly immunogenic and are FDA-approved for treatment with ICIs, agnostic to tumor type [40]. However, to date, the majority of research interrogating neoantigen-specific anti-tumor immunity has focused on SNV-specific T cells.

Abundant data from a variety of methodologies demonstrated that a high TMB predicted responses to ICIs in several cancer types. First, high-TMB tumor types (e.g., lung, melanoma, and bladder) responded better to immunotherapy. A study comparing the average TMB of various tumor types to the average ICI response rate in those tumor types found that TMB explained approximately 55% of the difference in response rates among the tumor types [41]. Additionally, individual tumors with a higher TMB within a given tumor type responded better to ICIs. For example, a multi-center, multi-ICI study of 1662 advanced cancer patients found that for NSCLC, melanoma, colorectal, bladder, and head and neck cancers, patients with tumors with a TMB in the top 20% for a given tumor type experienced significantly improved overall survival compared to those with a TMB in the bottom 80% [42]. This result was not due to the intrinsically better survival of patients with a high TMB, as patients with a high TMB on non-ICI therapies failed to experience improved survival [42]. Interestingly, the 20% cutoff for each tumor type resulted in dramatically different TMB per megabase (Mb) values among the tumor types. For example, the top 20% cutoff was 52 mutations/Mb for colorectal cancer versus 13.8 mutations/Mb for NSCLC, implying that each tumor type might require different TMB cutoffs [42]. Finally, multiple clinical trials have repeatedly shown that responses to ICI were associated with increased TMB in NSCLC, melanoma, bladder, colorectal, and several other cancer types [43,44,45,46,47,48]. These data led the FDA to approve Pembrolizumab (anti-PD-1) for advanced tumors with TMB greater than ten mutations/Mb, agnostic to tumor site [47,49], the only such tumor-type agnostic treatment to date.

Although a high TMB predicted response to ICI in multiple tumor types, RCC was an outlier. In studies comparing immunotherapy response between high- and low-TMB groups, RCC tumors responded well to ICI, yet the TMB values were low compared to those of 100 different tumor types [41,50]. In contrast to melanoma, CRC, NSCLC, bladder cancer, and head and neck cancer, RCC tumors with a TMB in the top 20% (>5.9/Mb) failed to show a significant improvement in OS compared to the bottom 80% [42,51]. Similarly, a study of 457 tumors found no significant difference in survival between the high- and low-TMB groups in the 57 RCC cases assessed [52]. Moreover, a study of 34 patients with metastatic RCC (mRCC) found no significant difference in TMB values between patients who had progressive disease (mean TMB = 3.01) and patients with complete response, partial response, or stable disease (mean TMB = 2.63) [53]. Finally, a similar lack of associations between TMB and response to ICIs in RCC was observed in the phase III CheckMate 214 trial [54], the phase 2 IMmotion 150 trial [55], and additional clinical trials of multiple ICIs [56,57] (see Table 1 for a list of phase III clinical trials and the main findings).

Given that neoantigens seem to make ideal targets for anti-tumor immunity, why does an increased TMB fail to predict successful immunotherapy in RCC? Studies suggest that indels may have an outsized effect on neoantigen load in RCC. For example, a large genomics study of 5777 tumors found that of 19 tumor types tested, RCC had the highest proportion of mutations comprising indels [39]. Furthermore, an epitope discovery study of six RCC patients found that indel mutation-specific T cells comprised up to 43% of the neoantigen-specific T cells [69]. Additionally, bioinformatic analyses of patient HLA-specific neoantigens from 335,594 nsSNVs and 19,849 frameshift indels found that frameshift indel mutations yielded approximately three times as many predicted neoantigens per mutation (2.00) compared to SNVs (0.64) [39]. The study further identified associations between response to ICIs and indel burden in RCC, which superseded the association with TMB. Thus, it is plausible that the number of neoantigens in RCC with high indel loads may be similar to the number of neoantigens in tumor types with increased SNVs. Supporting these findings, a study of 457 tumors found that TMB was associated with response to ICIs in melanoma and NSCLC, whereas indel burden was associated with response to ICIs in RCC [52]. However, studies examining tumors from the CheckMate-009, CheckMate-010, CheckMate-025, and CheckMate-214 clinical trials found that somatic copy number alterations, non-synonymous SNVs, predicted neoantigens, and frameshift indels failed to predict response to anti-PD-1 therapy in RCC [54,70]. To date, conflicting evidence implicates indel burden and response to ICIs in RCC, and further validation is required to use indel burden as a predictive biomarker in RCC.

## 4. Cytotoxic T Cells

As the predominant anti-tumor immune cell and the primary cell type that immune checkpoint blockade activates, abundant studies have investigated cytotoxic T cells and their effects on ICI response. Indeed, an analysis of 300 studies involving 70,000 individual tumors found that increased intratumoral CTLs were overwhelmingly associated with improved outcomes in most cancer types [71]. These observations led to the development of the “Immunoscore”, which assessed the level of memory CTLs at the invasive margin and was more prognostic than was the TNM stage for colorectal carcinoma [72,73]. Among several cancer types, increased CTL infiltration was not only prognostic, but also predictive of improved response to immunotherapy. For example, a meta-analysis of 33 clinical trials of ICIs found that patients with increased CTL experienced improved ORR (OR = 4.08, 95% CI 2.73–6.10), OS (HR = 0.52, 95% CI 0.41–0.67), and PFS (HR = 0.52, 95% CI 0.40–0.67) [74]. Moreover, a second meta-analysis of nine ICI clinical trials found that increased memory CTLs were associated with longer OS (HR 0.37, 95% CI 0.21–0.65) and PFS (HR 0.64, 95% CI 0.53–0.78) [75]. Furthermore, a meta-analysis investigating whole-genome and transcriptome sequencing of over 1000 patients among various phase III ICI clinical trials found a strong association between CD8 expression and response to ICI therapy [76]. These studies provide overwhelming evidence supporting CTL abundance as a predictive and prognostic biomarker for ICI in many tumor types.

The association between increased CTLs and response to ICI is mechanistically rational and clinically demonstrated for most tumor types. Surprisingly, increased CTLs in RCC repeatedly failed to predict response to checkpoint inhibition [71,77]. Interestingly, RNA-seq data from thousands of tumors among 18 tumor types found that CTLs infiltrate RCC tumors more than any other tumor type examined [78]. However, the effect of these intratumoral CTLs is ambiguous. For example, a study of 592 advanced RCC cases from the CheckMate 009, 010, and 025 clinical trials categorized tumors into “immune infiltrated” (73% of tumors), “immune excluded” (5% of tumors), and “immune desert” (22% of tumors) based on high numbers of CTLs in the stroma and epithelium, or absent CTLs, respectively. In this study, the immune category failed to predict OS or PFS after the use of ICIs [70]. Additionally, the JAVELIN renal 101 clinical trial found that increased intratumoral CTLs failed to predict PFS in patients receiving ICI [79]. Moreover, a clinical study of 24 RCC patients treated with dual checkpoint inhibition found that the response failed to associate with CTL infiltration; however, increased CTLs that expressed CD137 (a costimulatory molecule) trended towards improved response to therapy [80]. These studies highlighted the need for the further characterization of intratumoral T cells to identify active and effective anti-tumor CTLs in RCC and further interrogate the association of CTL subsets with response to ICIs.

## 5. T Cell Checkpoint Molecules

CTLs possess a unique capacity to kill cells of the body; thus, evolutionary pressure developed failsafe mechanisms to prevent overwhelming autoimmunity [81]. Once activated, CTLs upregulate PD1 [82], and if CTLs recurrently bind cognate peptide–MCH, they progressively upregulate additional inhibitory molecules such as CTLA-4, TIM3, LAG3, TIGIT, and CD39 [83,84,85,86,87]. Thus, if CTLs are chronically stimulated, T cell activity is inhibited when they bind to ligands expressed on antigen-presenting cells or tumor cells, thereby limiting out-of-control anti-self-immunity (autoimmunity) (Figure 2). However, these same mechanisms inhibit desirable CTL activity against the tumor, meaning inhibitory molecules are detrimental to anti-tumor immunity. Interestingly, CTLs that express multiple inhibitory checkpoint molecules within the tumor microenvironment are neoantigen-specific, tumor-specific, oligoclonal, and terminally exhausted [88,89,90]. Therefore, increased concentrations of terminally exhausted CTLs indicate that CTLs have the ability to access the tumor and recognize tumor cells. Reversing terminal exhaustion might enable these anti-tumor T cells to eliminate the tumor. Clinical studies show terminally exhausted T cells associated with positive response to ICIs in gastric [91], biliary tract [92], and non-small cell lung carcinomas [93]. Clinical trials blocking these additional checkpoint molecules have shown positive results in multiple tumor types [94,95,96].

Once again, RCC seems to be an outlier to the paradigm of increased numbers of exhausted CTLs and improved response to ICIs. PD1+ TIM3+ LAG3+ intratumoral CTLs in localized RCC were found to be polyclonal, dysfunctional, and were associated with lower cytotoxic effector granule expression and worse survival compared to RCC tumors containing TIM3- LAG3- CTL [97]. Additional studies found that CTLs with high levels of exhaustion markers expressed fewer effector cytokines and were associated with increased Tregs and M2 macrophages and decreased survival in RCC [98,99,100]. Given the polyclonal nature of the exhausted T cells and their association with an immunosuppressive microenvironment, these T cells may represent bystander T cells that may not participate in the anti-tumor immune response. Supporting this hypothesis, analyses of the CheckMate-010 and Checkmate-025 trials determined that tumors containing increased PD1+ CTL but not PD1+ TIM3+ LAG3+ CTL expressed increased effector cytokines and were correlated with improved PFS [101,102]. Ultimately, further investigation is required to determine the predictive capacity of various checkpoint inhibition molecules for response to immune checkpoint inhibition, and to date, no large clinical trials investigating TIM3 or LAG3 blockage have been reported for RCC.

## 6. Cytokines

Cytokines play a critical role in anti-tumor immunity. IL-2, TNF-α, IL-12 and IFN-γ are key effector cytokines expressed by TH1 cells, CTL, and M1 macrophages. These cytokines lead to tumor cell senescence and the potentiation of the anti-tumor effector function. Tumors recurrently mutate IFN receptor pathways as a means of immune escape, highlighting the importance of these cytokines in anti-tumor immunity [103]. Indeed, treatments of metastatic RCC with exogenously administered TNF-α and IL-2 were among the first anti-tumor immune therapies that were FDA-approved. These trials were met with some success and even provided cures in a small percentage of patients [104,105]; however, toxicities limited the use of these agents [106]. As predictive biomarkers, several studies investigating cytokines in ICI clinical trials have shown that a strong IFN-γ and/or TNF-α RNA signature was associated with improved response to ICIs in many cancer types, including RCC [107,108]. In addition to cytokines, chemokines recruit different cell types into the TME, thereby affecting the cellular composition of the tumor (reviewed [109]). For example, the chemokines CXCL9, CXCL10, and CXCL11 recruit CTL and TH1 cells into the TME and have been associated with improved survival in RCC [109,110]. Nivolumab was found to upregulate CXCL9 in metastatic RCC [111], and a transcriptome study of >1000 patients found that CXCL9 was the second-most important predictor of response to ICIs in multiple tumor types, including RCC [76].

In contrast to the beneficial, pro-inflammatory activity of the cytokines above, cytokines expressed by tumor cells, M2 macrophages, myeloid-derived suppressor cells (MDSC), and Tregs inhibit the anti-tumor immune response. For example, TGF-β and IL-10 secreted by M2 macrophages, MDSCs, and tumor cells inhibit CTL activity and enhance TH cell conversion into Tregs [15]. Tregs also secrete TGF-β and IL-10 that further inhibit anti-tumor CTL [14,15,112]. Some chemokines recruit immune-inhibiting cells into the microenvironment. For example, IL-8 promotes angiogenesis and recruits MDSCs and neutrophils into the TME, resulting in decreased anti-tumor immunity [113]. Similarly, CCL17 and CCL22 secreted by M2 macrophages and tumor cells recruit Tregs to the TME, thereby inhibiting anti-tumor immunity [114,115]. Cytokines such as these have been studied as biomarkers in clinical trials. Checkmate-025 patients with higher IL-8 in peripheral blood responded poorly to Nivolumab (HR: 2.56, 95% CI: 1.89–3.45) compared to patients with lower levels of IL-8 [116]. Furthermore, analysis of the Immotion150 cohort found increased IL-8 in serum and increased IL-8 RNA expression in circulating monocytes associated with decreased antigen presentation and poor response to therapy [117]. RCC patients with an increased plasma CCL17/CCL22 ratio had decreased overall survival [114], and patients with increased expression of the receptor for these cytokines (CCR4) had worse overall survival in multiple tumor types [118,119], including RCC [120].

Although chemokines and cytokines play critical roles in the anti-tumor immune response and often correlate with survival, in practice, cytokine and chemokines are challenging to adapt to the pathological assessment of tumors. Cytokines are expressed on a continuum, and semi-quantitative assays such as IHC are difficult to modulate for a correct assessment of small differences in protein expression. However, cytokine genes carry a lot of weight in gene expression profile (GEP) panels (discussed below) and have potential for improving predictions of ICI benefit.

## 7. Angiogenesis Factors

Von Hippel–Lindau (VHL) is a tumor suppressor that leads to the ubiquitination and destruction of hypoxia-inducible factor (HIF)1α and HIF2α (HIF1/2α) in normoxic conditions and is mutated in 64% of ccRCC cases [121]. In normal cells under normoxic conditions, the HIF1/2α proteins are hydroxylated, allowing recognition by VHL, part of an E3 ubiquitin ligase complex. This leads to the ubiquitin-mediated proteasomal destruction of HIF1/2α. In hypoxic conditions, HIF1/2α proteins are not hydroxylated, thereby preventing VHL recognition and leading to stable HIF1/2α. Stabilized HIF1/2α migrates to the nucleus, dimerizes with HIFβ proteins and acts as transcription factors for several tumor-promoting angiogenic, mitogenic, and erythropoietic proteins. VHL loss causes the stabilization of HIF1/2α under normoxic conditions leading to increased expression of the tumor-promoting proteins normally expressed under hypoxic conditions (Figure 3) (reviewed in [122,123,124,125]). Some of the key immune inhibiting proteins that are expressed when VHL is lost include PD-L1 (discussed below), vascular endothelial growth factor (VEGF) (discussed below), platelet-derived growth factor (PDGF), and IL8 (discussed above) [124,126]. Several studies have investigated VHL mutation status, as well as several resultant effector proteins, as biomarkers for ICIs in RCC.

HIF-induced vascular endothelial growth factor (VEGF) family cytokines and their receptors (VEGFR) deserve special mention, as they play critical roles as drivers of angiogenesis and immune suppression in RCC and are the target of several therapies in RCC. VEGF signaling recruits MDSCs [127], Tregs [128], and neutrophils, causes macrophage polarization to the M2 phenotype [129], decreases the cytotoxicity of NK and T cells, and decreases the recruitment of CTLs [130,131]. VEGFα also induces haphazard and poorly developed neovascularization, with reduced capacity for anti-tumor immune cell extravasation [132,133]. Antibodies blocking VEGF or VEGFR work synergistically with ICIs to improve anti-tumor activity in vitro, and in vivo [134,135]. These inhibitors were first approved for metastatic RCC in combination with TNFa [136,137] and, subsequently, as monotherapies replacing cytokine therapy [138,139,140,141,142]. Recently, combinations of VEGF/VEGFR inhibitors with ICIs showed benefits in phase III trials [59,60,65].

Given the critical roles that VHL and VEGF pathways play in RCC tumorigenesis and anti-tumor immunity, studies have been surprisingly ambiguous in terms of the utility of these biomarkers for predicting response to ICIs. For example, a study of 34 patients with metastatic RCC found that VHL mutations failed to associate with responses to ICIs [53]. Similarly, VHL mutation status failed to associate with clinical benefit in a whole-exome sequencing study of 35 patients with metastatic ccRCC treated with Nivolumab [143]. In contrast, a retrospective analysis of transcriptome data from the CheckMate 025 clinical trial found high HIF1α levels associated with longer PFS (*p* = 0.0249) and OS (*p* = 0.0014) in patients treated with ICIs [144]. Given the ambiguity of VHL mutation status versus response to ICIs, one study looked at serum levels of the effector cytokine VEGFα and identified an association between increased VEGFα and poor response to ICIs in melanoma [145]. However, the variable expression levels make the assessment of these markers in tumors challenging with semi-quantitative IHC techniques, and RNA expression analyses are typically required, as discussed below.

## 8. Gene Expression Profiles

Given the complexity of the cytokine, chemokine, angiogenesis, and cellular networks that operate in the anti-RCC immune response, a combined measurement of these different response effectors may improve predictions for response to ICIs. Various groups investigated gene expression profiles (GEP) using transcriptome sequencing data to predict response to ICIs. Interrogation of the phase II Immotion150 clinical trial cohort identified a gene expression panel for an effective T cell response (Teff) that included transcripts for T cell presence and response, IFN-γ-related genes, checkpoint inhibitor genes, and antigen presentation genes. Patients with Teff-high signatures in the immunotherapy arms experienced improved ORR (49% vs. 16%) and improved PFS (HR 0.50; 95% CI, 0.30–0.86) [55]. Furthermore, a study analyzing the JAVELIN Renal 101 phase III trial cohort found that patients with an increased expression of 26 immune related genes in the avelumab plus axitinib arm experienced longer PFS than those with low expression levels (HR 0.60; 95% CI 0.439–0.834), but this association was absent in the Sunitinib arm [79]. A validation dataset from the phase II trial of the same agents showed similar results [79]. However, these GEPs failed to identify an association with survival in the Checkpoint-214 phase III clinical trial [54], implying that different immunotherapy regimens may require unique GEPs [146]. Ultimately, GEPs have cost and availability constraints that make clinical applications challenging. Questions remain about the utility of GEP to predict responses to ICIs, requiring further validation of this exciting field.

## 9. Programmed Death-Ligand 1

Programmed death-ligand 1 (PD-L1) is the most thoroughly studied biomarker for cancer ICI therapy [147]. PD-L1 expression on tumor cells and tumor-infiltrating immune cells [148] interacts with PD-1 on activated T cells, precipitating T cell death, reducing cytokine production, and inhibiting anti-tumor activity [149]. Unfortunately, several technical difficulties make assessing PD-L1 expression challenging in practice, and comparison between trials is fraught with uncertainty. For example, some studies measure PD-L1 expression on tumor cells alone using the Tumor Proportion Score (TPS) methodology [150], while other studies assess PD-L1 expression on both tumor and immune cells using the Combined Positivity Score (CPS) methodology [150]. Moreover, different studies use various PD-L1 expression levels to determine overall positivity, such as 1%, 5%, 10%, 25%, and 50% [151]. Finally, geographic and temporal heterogeneity within a tumor make PD-L1 expression positivity prone to sampling bias [151]. These challenges make PD-L1 expression difficult to interpret and compare among various clinical trials.

In addition to the technical challenges of PD-L1 assessment, the theoretical rationale for PD-L1 as a biomarker is weakened due to the multiple mechanisms by which PD-L1 expression is induced. For example, in some cases, IFN-γ released by tumor-infiltrating T cells induces the upregulation of PD-L1 on tumor cells [152]. In these cases, tumor cell PD-L1 expression indicates the presence of infiltrating activated T cells and the possible susceptibility of the tumor to immune attack [152]. In contrast, activated oncogenic signaling pathways and hypoxia also induce PD-L1 expression in tumor cells [153]. For example, constitutive PI3K–mTOR pathway oncogenic signaling due to PTEN loss [154] and excessive HIF signaling due to VHL loss [155] both lead to increased PD-L1 expression. In such cases, upregulated PD-L1 expression is unrelated to active anti-tumor immunity, but rather, is a byproduct of oncogenic signaling; therefore, in these cases, upregulated PD-L1 is unlikely to indicate tumors that are susceptible to ICI-meditated anti-tumor immunity. The competing mechanistic underpinnings of PD-L1 expression make it difficult to understand the implications of changes in expression levels, and help to explain the different predictive power of PD-L1 expression among different tumor types.

Despite the inherent challenges of using PD-L1 as a marker for successful immunotherapy, it remains one of the few biomarkers to direct ICIs use in some tumor types. PD-L1 expression has been shown to predict response to various ICIs in head and neck squamous cell carcinoma, triple-negative breast cancer, urothelial carcinoma, and gastric carcinoma [156]. In some health jurisdictions, increased PD-L1 expression is a pre-requisite for initiating therapy in NSCLC and head and neck squamous cell carcinoma [156,157]. However, in other cancer types, including melanoma, hepatocellular carcinoma, colorectal carcinoma, endometrial carcinoma, and ccRCC, PD-L1 expression levels are not used to direct treatments with ICIs.

Studies in RCC demonstrate contradicting associations between PD-L1 expression and responses to ICIs. In the CheckMate 214 phase III clinical trial, both PD-L1-high tumors (≥1% PD-L1) and PD-L1-low tumors (<1% PD-L1) responded better to nivolumab plus ipilimumab compared to sunitunib; however, the PD-L1-high tumors had a better average response than the PD-L1-low tumors did [158]. Setting an arbitrary PD-L1 expression level necessary for ICI therapy in this trial would have denied patients access to potentially beneficial therapy. Remarkably, CheckMate-025 phase III patients with tumors with PD-L1 expression of ≥1% experienced significantly decreased overall survival compared to patients with PD-L1 expression of <1% (21.8 months (95% CI, 16.5–28.1) vs. 27.4 months (95% CI, 21.4—not estimable) [66]; however, both groups experienced improved survival compared to the everolimus arm [66]. Furthermore, the phase III trial Checkmate 9ER showed improved PFS and OS in patients treated with ICIs irrespective of PD-L1 expression levels [67]. Moreover, the phase III clinical trials CLEAR and KEYNOTE-426 demonstrated improved overall survival for patients receiving ICIs without significant differences between PD-L1-high and -low groups [68,159]. Meta-analyses of these and other ICI clinical trials in RCC, including 4635 patients treated with a variety of checkpoint inhibitors, showed that overall survival failed to associate with PD-L1 expression levels [160]. These findings demonstrate the limited utility of PD-L1 as a predictive biomarker in RCC, and treatment decision-making should not rely on PD-L1 expression levels in RCC.

Currently, both TPS and CPS scoring systems specifically avoid including PD-L1 expression on vasculature, yet PD-L1 is highly expressed on both human and murine lymphatic endothelial cells and may inhibit anti-tumor immunity [161,162]). A study of mouse models of melanoma found that knocking out PD-L1 specifically in the vasculature resulted in improved extravasation of CTLs into the TME and tumor control, indicating that PD-L1 expression on vessels resulted in immune escape [163]. Further, PD-L1 expression on lymphatic endothelial cells specifically induced Treg extravasation into the TME of mice [162]. In humans, PD-L1 expression on vascular endothelial cells was associated with decreased intratumoral CTLs in hepatocellular carcinoma [164]. These studies provide an impetus for additional research to assess the effects of PD-L1 expression on RCC vasculature and may help improve the predictive power of PD-L1 for ICI response.

## 10. PBRM-1 Loss

Polybromo-1 (PBRM1) encodes the DNA-targeting protein in the PBAF SWI/SNF chromatin remodeling complex and suppresses the hypoxia transcription pathways induced by VHL loss [165]. Loss of heterozygosity at chromosome 3p occurs in 91% of ccRCC, deleting one copy of both VHL and PBRM-1 [166], and recurrent second hits to both VHL and PBRM-1 are the two most common gene losses in ccRCC [166,167]. A study comparing RNA-seq data from PBRM-1-deficient and PBRM-1-proficient cell lines found two major pathways upregulated in PBRM-1-deficient tumors, with contrasting theoretical effects on anti-tumor immunity [143,168]. On one hand, PBRM-1 deficiency increased hypoxia-related transcripts, thereby theoretically inhibiting anti-tumor immunity (as explained above). On the other hand, PBRM-1-deficient tumors expressed increased cytokine pathway related genes such as IFN-γ, IL-2, IL-12, and CCL21, theoretically enhancing anti-tumor immunity. These contrasting theoretical effects of PBRM-1 loss on anti-tumor immunity provide the rationale for studying its use as a biomarker for ICIs.

Several studies have interrogated PBRM-1 as a predictive biomarker for response to ICIs. For instance, a whole-exome sequencing study of nivolumab-treated ccRCC cases found that patients experiencing clinical benefit from ICI were more likely to harbor PBRM-1 mutations (9/11 patients) compared to patients who experienced no clinical benefits (3/13 patients) [143]. A second cohort treated with ICI monotherapy or combination ICI therapies found similar improved clinical benefits in patients with PBRM-1 loss (17/27 patients with mutation) compared to patients with no benefits (4/19 patients with mutation) [143]. Similarly, an analysis of a subset of CheckMate-009 patients [169] and a pooled analysis of three clinical trial patients [70] found that PBRM-1 truncation mutations were associated with improved OS (HR = 0.65, 95% CI = 0.44–0.96, and *p*  =  0.03, and *p* < 0.001, respectively). These studies provide promise for PBRM-1 mutation status as a biomarker of response to ICIs, and a validation of these findings may lead to improved patient selection for ICIs. However, in all cases, some patients without PBRM-1 mutations responded well to ICIs, and caution is required prior to utilizing this marker in clinical decision-making.

## 11. Future Directions

ICI biomarkers useful in many tumor types conspicuously fail to predict improved response to ICIs in RCC. It is likely that these biomarkers predict improved responses in only subsets of RCC patients and competing mechanistic oncogenic drivers in RCC make assessing single biomarkers ineffective. For example, tumor PD-L1 expression in cases with increased CTLs may indicate active anti-tumor immunity, while tumor PD-L1 expression in cases with increased angiogenesis markers may indicate resistance to ICIs. Thus, combinations of biomarkers with spatial resolution may identify unique cell patterns for predicting response to ICIs. New high-dimensional imaging technologies that can measure hundreds or thousands of markers on each cell on a slide are beginning to detangle the various mechanisms involved in anti-tumor immunity [170,171]. Using a combination of single-cell RNA sequencing, high-dimensional FISH, high-dimensional protein immunofluorescence, and on-tissue transcriptomics, an atlas of human breast has been generated, whereby all cells in human breast tissues and their spatial relationships have been fully characterized [172]. When applied to cancer, these types of studies reveal unique cell–cell interactions that provide predictive information unavailable through either bulk transcriptomics or single-stained IHC [171]. For example, one study of cutaneous T cell lymphoma simultaneously measured 56 immunofluorescence markers and identified 21 different cell types and states [173]. The authors identified a unique association between response to anti-PD1 therapy and the average distance between CD4+ PD1+ T cells and Tregs [173]. Encouragingly for biomarker discovery, additional experiments would require only three markers to validate this association.

Due to the complex nature of sample preparation and imaging, these technologies are unlikely to be used in the clinic within the foreseeable future. However, the experimental cell–cell interactions that these studies identify may lead to clinically actionable low-dimensional IHC biomarkers. Current efforts are underway to standardize and validate multiplex immunofluorescence biomarkers with promising levels of reproducibility [174]. Eventually, these technologies will unravel the complex dynamics of cell–cell patterns influencing anti-tumor immunity and generate new biomarkers for ICI.

## 12. Conclusions

Cancer immunotherapy has revolutionized the treatment of advanced renal cell carcinoma, providing hope to patients previously considered palliative. Unfortunately, with these new treatments come adverse side effects making patient selection for therapy challenging. Identifying biomarkers that predict successful immunotherapy in RCC would help to alleviate unnecessary suffering and improve patient outcomes. Unfortunately, to date, sufficiently validated biomarkers are conspicuously absent. RCC is an outlier in that TMB, tumor-infiltrating lymphocytes, and PD-L1 expression fail to predict response to ICIs. Indeed, the unique mechanistic underpinnings of RCC initiation, progression, and immune evasion require disease-specific research to identify novel biomarkers for improved ICI delivery.

## Figures and Tables

**Figure 1 jcm-12-04987-f001:**
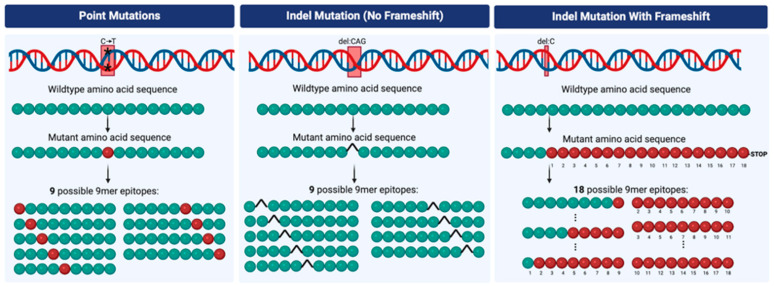
Potential epitopes from different types of mutations. In each panel, DNA is depicted on the top row with a specific type of mutation. The green circles below represent the wildtype amino acids resulting from the translation of the mutant and wildtype DNA sequences. The red circles represent mutated amino acids translated from the mutated DNA. The 9mer epitopes represent all possible 9mer amino acid sequences that contain a mutation, potentially forming a neoantigen. Indel mutations with frameshifts result in sequences of mutated amino acids that continue until premature stop codons are translated. In this example, 18 possible 9mer epitopes were generated from a single frameshift mutation (del:C). del:CAG = deletion of 3 nucleotides; del:C = deletion of one nucleotide; C→T with * represent a point mutation. Created with Biorender.com accessed on 1 January 2023.

**Figure 2 jcm-12-04987-f002:**
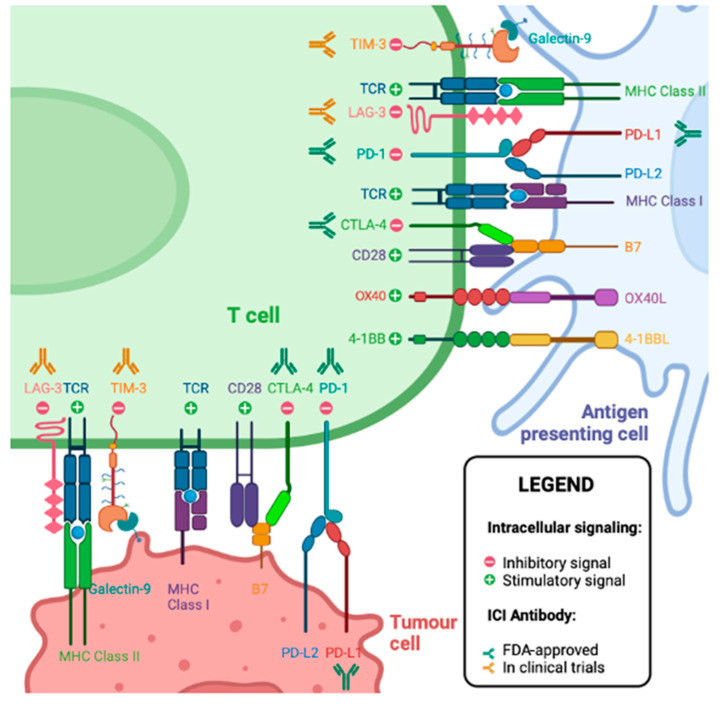
T cell-stimulatory and -inhibitory receptors and ligands. T cells (green cell) may become activated if the T cell receptor (TCR) recognizes and binds a peptide–MHC complex on a dendritic cell (blue cell). However, the costimulatory and inhibitory ligands expressed by the dendritic cell have a profound effect on T cell activation. If the inhibitory ligands on APCs bind inhibitory molecules on T cells (red circle with “−”), the T cell may undergo anergy or cell death. If costimulatory ligands on dendritic cells bind costimulatory molecules on a T cell (green circles with “+”), the T cell may rapidly proliferate and differentiate into anti-tumor cytotoxic T cells (CTL). Once an activated CTL binds its cognate peptide–MHC on a tumor cell (brown cell) it may kill the tumor cell. However, tumor cells may also express inhibitory ligands that inhibit T cell activity when bound to inhibitory molecules on T cells. Blocking this interaction with ICIs releases the breaks on T cell-mediated anti-tumor immunity. ICI antibodies approved in RCC are in green. ICI antibodies under investigation for RCC are in orange. Created with Biorender.com, accessed on 1 January 2023.

**Figure 3 jcm-12-04987-f003:**
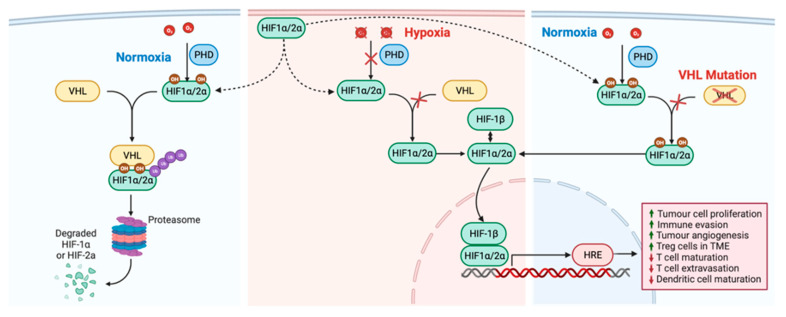
VHL loss leading to HIF stabilization under normoxic conditions. Under normoxic conditions, HIF1/2α is hydroxylated. VHL recognizes hydroxylated HIF1/2α, leading to ubiquitination and proteasome-mediated degradation. Under hypoxic conditions, HIF1/2α is not hydroxylated, disallowing VHL from binding. HIF1/2α is thereby stabilized, binds with HIFβ, and transcribes pro-tumor factors important in various immune-inhibiting and angiogenesis pathways. When VHL is lost through mutations, HIF1/2α is stabilized under normoxic conditions, leading to the pathological transcription of pro-tumorigenesis factors. Arrows indicate movement of molecule; crossed out arrows indicate lost movement of molecule. Created with Biorender.com, accessed on 1 January 2023.

**Table 1 jcm-12-04987-t001:** List of phase III clinical trials of ICIs in RCC and significant findings (OS = overall survival; PFS = progression-free survival; HR = hazard ratio; NS = not significant; NR = not reached; mo = months).

Trial Name	Trial Features	Treatment Arm (s)	Control Arm	Significant Findings (Treatment vs. Control)
Immotion 151 [58,59]	Phase III, multicenter RCT on previously untreated metastatic RCC	atezolizumab + bevacizumab	sunitinib	OS PD-L1+: 38.7 mo vs. 31.6 mo PFS: PD-L1+: 11.2 mo vs. 7.7 mo (HR = 0.74)
JAVELIN Renal 101 [60,61]	Phase III clinical trial of previously untreated advanced-RCC patients	avelumab + axitinb	sunitinib	OS PD-L1+: NR vs. 36.2 mo (HR = 0.81) PFS PD-L1+: 13.9 mo vs. 8.2 mo (HR = 0.58)
CheckMate 214 [62,63,64]	Phase III clinical trial of previously untreated advanced-ccRCC patients	nivolumab + ipilumumab	sunitinib	OS PD-L1+: 66.8 mo vs. 23.9 mo (HR = 0.57) OS PD-L1−: 59.2 mo vs. 41.9 mo (HR = 0.77) PFS PD-L1+: NR vs. 5.6 mo PFS PD-L1−: 9 mo vs. 5.4 mo
KEYNOTE-426 [65]	Phase III clinical trial on previously untreated advanced-ccRCC patients	pembrolizumab + axitinib	sunitinib	OS PD-L1+: HR = 0.54 OS PD-L1−: HR = 0.59, NS PFS PD-L+: 15.3 mo vs. 8.9 mo (HR = 0.62) PFS PD-L1−: 15 mo vs. 12.5 mo (HR = 0.87, NS)
CheckMate 025 [66]	Phase III clinical trial on advanced ccRCC patients previously on antiangiogenic therapy	nivolumab	everolimus	OS PD-L1+: 21.8 mo vs. 18.8 mo (HR = 0.79)OS PD-L1−: 27.4 mo vs. 21.2 mo (HR = 0.77)PFS PD-L1+: 4.6 mo vs. 4.4 mo (HR = 0.88)
CheckMate 9ER [67]	Phase III clinical trial on previously untreated advanced-ccRCC patients	nivolumab + cabazantinib	sunitinib	OS PD-L1+: HR = 0.80 PFS PD-L1+: HR = 0.49
CLEAR [68]	Phase III clinical trial on systemic therapy naïve patients with advanced RCC	pembrolizumab + lenvatinib; lenvatinib + everoliums	sunitinib	PFS PD-L1+; P+L vs. S: HR = 0.40 PFS PD-L1−, P+L vs. S: HR = 0.39

## Data Availability

No new data were created in this review.

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
