# Peer review of "Biomarkers for Immune Checkpoint Inhibitors in Renal Cell Carcinoma"

_jcm, 2023, doi:10.3390/jcm12154987_

Round 1
Reviewer 1 Report
The review paper is well-written and provides comprehensive overview of the current acknowledge of immunotherapy, in particular in renal cell carcinoma. Only a minor error is found. In Line 41, "oncocytoma" should be deleted, since the author is talking about renal cell carcinoma in this sentence.
Author Response
Dear editor and reviewers,
Thank you for the very quick turn around time for reviewing our paper entitled “Biomarkers for Immune Checkpoint Inhibitors in Renal Cell Carcinoma”. We agree with all reviewers that ours is a very comprehensive overview of the current knowledge of biomarkers for immune checkpoint inhibition in renal cell carcinoma.
The information we convey in our review is accessible in some studies published to date and may be known to an expert in renal cell carcinoma immunotherapy; however, we feel that our study is unique as it provides a combination of mechanistic rationale in addition to clinical experience for each biomarker we have examined. This combination is lacking in the literature and will be a great resource for the general clinician scientist that comprise JCM reader base. We anticipate that our highly readable work will be well cited and summarizes the current knowledge of biomarkers in RCC.
We have attempted to break down the narrative reviews into point form requests for amendments. We have included point-by-point responses to the reviewers’ comments.
Itemized list of issues to resolve. We have marked our responses in bold:
Reviewer 1:
- “"oncocytoma" should be deleted”.
Answer: “Oncocytoma” has been removed from line 41.
Reviewer 2:
- The introduction would benefit from a clearer statement of the study's aims and objectives, more information on specific indications and their prevalence, and the inclusion of more recent literature to contextualize the current state of knowledge and the knowledge gap the study seeks to fill.
Answer: Thank you for pointing out improvements possible in our introduction. We have added the current guidelines for use of ICI in RCC into the introduction in lines 60 – 67. We additionally included citations including references of work presented at the most recent ASCO meeting (references 8 – 11).
We have substantially edited the final paragraph in the introduction to better characterize our review aims and outline our goals. These can be found in lines 72 – 83.
- Choose those tests that are validated or in the process of being validated.
Answer: All biomarkers examined are in various stages of validation or experimentation.
- Keep acronyms consistent
Answer: We have identified PDL1 (vs PD-L1) as inconsistently in our manuscript and have corrected and standardized this acronym to PD-L1.
- Correct any typos
Answer: We have corrected minor typos scattered in the paper
Reviewer 3:
- The authors should strive to introduce new concepts and highlight potential future directions that will attract and engage readers.
Answer: Thank you for encouraging us to expand our research into biomarker discovery that may be on the horizon. We have added a paragraph in the PD-L1 section that discussed PD-L1 expression on vasculature (which is currently overlooked in PD-L1 scoring). These additions can be found in lines 487 – 497.
In addition, we have added a new section entitled Future Directions, in which we discuss the potential of high-dimensional multiplex tissue imaging and cell – cell neighborhood analysis to add to biomarker discovery. These changes can be found in lines 527 - 555.
- Figures and tables presented in the paper closely resemble those found in other published works, further diminishing the originality of the review.
Answer: Although one may find similar figures in other reviews, the figures are fundamental to understanding our review. They are necessary for a general clinician scientist reader to follow our paper. We believe that it would be difficult to expect a reader to search for other figures in other papers to contextualize our review. We feel that the high quality images in our manuscript will generate increased interest in our review and lead to increased frequency of citations.
The table is a list of phase III clinical trials that is current and up to date. We feel that it is a useful resource reference for this review and helps the reader itemize current clinical trial results.
Reviewer 2 Report
It is true that the topic chosen for the review is highlighted, since in recent years there has been a growing interest in the search for and identification of biomarkers through less invasive techniques that ensure better management of the disease and improve the quality of patient health, especially with regard to genitourinary cancers, which are among the most expensive for health systems and are based on invasive techniques for monitoring patients every few months and for years, thus worsening both their mental and physical health. The authors reported a review trying to assess potential biomarkers that predict responses to immune checkpoint inhibitors, focusing on ccRCC. For each potential biomarker, They presented the mechanistic rationale underpinning the utility of the biomarker and provide evidence for its use in ccRCC and other types of cancer. The manuscript needs some amendments before consideration for publication.
- The introduction would benefit from a clearer statement of the study's aims and objectives, more information on specific indications and their prevalence, and the inclusion of more recent literature to contextualize the current state of knowledge and the knowledge gap the study seeks to fill.
- The workflow of the study has no interest beyond describing it in materials and methods. After the screening criteria they have chosen, it is more relevant to choose those tests that are validated or in the process of being validated, since what is sought is real biomarkers that can be used in the clinic. Here is a reference to some examples: PMID: 37189689 ; PMID: 37109725
- Sometimes they use different acronyms to define the same thing.
- Check typos.
Author Response

(The authors gave the same response as above.)

Reviewer 3 Report
In their article, Martin et al. examine various biomarkers for predicting the response to immune checkpoint inhibitors in renal cell carcinoma. The topic of biomarkers and their association with treatment response has been extensively explored in the literature, and this review paper adds to the existing body of knowledge. However, it does not offer significant new insights or novel concepts to engage readers.
The authors primarily discuss well-known issues and reiterate points that have already been extensively addressed in previously published works. Consequently, the review lacks the fresh perspective needed to captivate readers and provide them with groundbreaking information.
Furthermore, the figures and tables presented in the paper closely resemble those found in other published works, further diminishing the originality of the review. To enhance the value of their work, the authors should have introduced innovative concepts or proposed future directions that would intrigue readers and encourage further research in the field.
Overall, while Martin et al.'s review provides a comprehensive overview of biomarkers for immune checkpoint inhibitors in renal cell carcinoma, it falls short in terms of offering novel insights or unique perspectives. To make it more compelling, the authors should strive to introduce new concepts and highlight potential future directions that will attract and engage readers.
Author Response

(The authors gave the same response as above.)

Round 2
Reviewer 2 Report
No more comments.
Author Response
Thank you very much!
Reviewer 3 Report
All responses from the authors are already published issues, not novelty ones showing the authors' originality. The authors need to provide brand-new insights which attract readers.
Author Response
Thank you again for reviewing our revised manuscript. We understand the reviewer’s concern. As we have explained, the information we convey in our review is accessible in some studies published to date and may be known to an expert in renal cell carcinoma immunotherapy; however, we feel that our study is unique as it provides a combination of mechanistic rationale in addition to clinical experience for each biomarker we have examined. This combination is lacking in the literature and will be a great resource for the general clinician scientist that comprise JCM reader base. As the reviewer suggested before, we added a paragraph in the PD-L1 section that discussed PD-L1 expression on the vasculature (currently overlooked in PD-L1 scoring). These additions can be found in lines 487 – 497. In addition, per the previous comments, we have added a new section entitled Future Directions, in which we discuss the potential of high-dimensional multiplex tissue imaging and cell–cell neighborhood analysis to add to biomarker discovery. These changes can be found in lines 527 - 555. The above future directions have not been well conducted or reported so far, and they are actually the current research focus of our group through multidisciplinary and international collaboration.